# New Findings on LMO7 Transcripts, Proteins and Regulatory Regions in Human and Vertebrate Model Organisms and the Intracellular Distribution in Skeletal Muscle Cells

**DOI:** 10.3390/ijms222312885

**Published:** 2021-11-28

**Authors:** Geyse Gomes, Mariana Juliani do Amaral, Kayo Moreira Bagri, Larissa Melo Vasconcellos, Marcius da Silva Almeida, Lúcia Elvira Alvares, Claudia Mermelstein

**Affiliations:** 1Instituto de Ciências Biomédicas, Universidade Federal do Rio de Janeiro, Rio de Janeiro 21941-901, Brazil; geyse@histo.ufrj.br (G.G.); kayobiomed14@histo.ufrj.br (K.M.B.); larissavasconcellos@ufrj.br (L.M.V.); 2Faculdade de Farmácia, Universidade Federal do Rio de Janeiro, Rio de Janeiro 21941-901, Brazil; marianajamaral@yahoo.com.br; 3Instituto de Bioquímica Médica Leopoldo de Meis, Universidade Federal do Rio de Janeiro, Rio de Janeiro 21941-901, Brazil; marcius_3@hotmail.com; 4Departamento de Bioquímica e Biologia Tecidual, Universidade de Campinas (UNICAMP), Campinas, São Paulo 13083-872, Brazil; lealvare@unicamp.br

**Keywords:** LMO7, skeletal muscle, zebrafish, chicken, intrinsically disordered proteins, bioinformatics, calponin homology, PDZ, LIM

## Abstract

LMO7 is a multifunctional PDZ–LIM protein that can interact with different molecular partners and is found in several intracellular locations. The aim of this work was to shed light on LMO7 evolution, alternative transcripts, protein structure and gene regulation through multiple in silico analyses. We also explored the intracellular distribution of the LMO7 protein in chicken and zebrafish embryonic skeletal muscle cells by means of confocal fluorescence microscopy. Our results revealed a single *LMO7* gene in mammals, sauropsids, *Xenopus* and in the holostean fish spotted gar while two *lmo7* genes (*lmo7a* and *lmo7b*) were identified in teleost fishes. In addition, several different transcripts were predicted for *LMO7* in human and in major vertebrate model organisms (mouse, chicken, *Xenopus* and zebrafish). Bioinformatics tools revealed several structural features of the LMO7 protein including intrinsically disordered regions. We found the LMO7 protein in multiple intracellular compartments in chicken and zebrafish skeletal muscle cells, such as membrane adhesion sites and the perinuclear region. Curiously, the LMO7 protein was detected within the nuclei of muscle cells in chicken but not in zebrafish. Our data showed that a conserved regulatory element may be related to muscle-specific *LMO7* expression. Our findings uncover new and important information about LMO7 and open new challenges to understanding how the diverse regulation, structure and distribution of this protein are integrated into highly complex vertebrate cellular milieux, such as skeletal muscle cells.

## 1. Introduction

LIM domain only protein 7 (LMO7) is a large protein (1683 residues) that orchestrates many protein–protein interactions. It contains a LIM domain (a unique cysteine-rich zinc-binding domain), a calponin homology (CH) domain, a PDZ domain and an F-box (FBX) domain [1]. Combining distinct functional domains in one protein, the PDZ–LIM proteins have been related to wide-ranging and multicompartmental cell functions during development and homeostasis. For instance, these proteins are known to mediate signaling between the nucleus and the cytoplasm, sequester nuclear factors to the cytoplasm and interact with actin microfilaments [2]. In addition, the PDZ–LIM proteins facilitate the assembly of protein complexes, regulate gene transcription, control mitosis progression, influence spindle assembly checkpoint besides being involved in the ciliary function [3,4,5,6,7]. Regarding its intracellular distribution, the LMO7 protein can be found in the nucleus, perinuclear region, cytoplasm and/or on cell surface, particularly in cell–cell adhesions (where it can interact with nectin and E-cadherin through afadin and alpha-actinin) and focal adhesions [2,8,9,10]. Importantly, misregulation of the PDZ–LIM proteins is involved in cancer [11,12].

LMO7 has been described as an important regulator of skeletal muscle- and heart-relevant genes [8]. Accordingly, LMO7 is expressed in somites and in important structures involved in heart development in different vertebrates, including zebrafish, chicken and mouse embryos [13]. In addition, deletion of *Lmo7* in mice causes a muscular dystrophy phenotype similar to Emery–Dreifuss muscular dystrophy (EDMD) [8]. LMO7 binds the emerin protein and is involved in the regulation of the emerin gene [8,14].

The LMO7 role in skeletal muscle structure and function is still controversial. While Lao et al. [15] demonstrated that *Lmo7*-null mice have normal skeletal muscles regarding morphology, physiological function and regeneration capabilities, Mull et al. [16] reported impaired skeletal muscle function in *Lmo7*-mutant mice involving growth retardation and decreased fiber size. Assuming that these differences cannot be attributed to the methodology used in these studies, the role of LMO7 in skeletal muscle cells remains to be further elucidated.

Using C2C12 mouse myoblast cell cultures, Dedeic et al. [17] reported that LMO7 is required for myoblast differentiation. *Lmo7*-downregulated myoblasts exhibited reduced expression levels of *Pax3*, *Pax7*, *Myf5* and *MyoD*, whereas *Lmo7* overexpression increased *MyoD* and *Myf5* expression. LMO7 was found in the cytoplasm of myotubes and within the nuclei of myoblasts [17]. Importantly, LMO7 was shown to bind to the *Pax3*, *MyoD* and *Myf5* promoters during myogenesis [17].

The role of LMO7 in skeletal muscles was further analyzed in chicken muscle cell cultures [10]. Knockdown of *LMO7* using siRNA induced a decrease in the number of myoblasts and in the myotube size. Activators of the Wnt/beta-catenin pathway (Wnt3a and BIO) were able to revert these effects, suggesting an interaction between the Wnt/beta-catenin and LMO7-mediated signaling pathways.

This work aimed to shed light on LMO7 evolution, alternative transcripts, protein structure and gene regulation through multiple in silico analyses to provide support for future studies in different biological contexts, including skeletal muscles. In addition, we analyzed the spatial distribution of the LMO7 protein in myogenic cells of chicken and zebrafish, two important model organisms used to investigate skeletal muscle development, growth, and repair.

## 2. Results and Discussion

### 2.1. Identification of LMO7 Orthologous Genes in Different Vertebrates

To identify the orthologs of the human *LMO7* gene in species representative of different vertebrate groups, we started by performing a systematic search in the Ensembl and Gene NCBI databases. A single *LMO7* gene was identified in the genome of placental and marsupial mammals (mouse and opossum), sauropsids (chicken, zebra finch, Chinese softshell turtle and anole lizards), amphibian (*Xenopus tropicalis*) and in the spotted gar, a holostean fish. In contrast, in teleost fishes (zebrafish and catfish), two *lmo7* genes (*lmo7a* and *lmo7b*) were identified (Appendix A).

To understand the evolutionary relationship between the LMO7 sequences identified, we performed a phylogenetic analysis based on the predicted amino acid sequences. The *Branchiostoma floridae* (cephalochordate) Lmo7 protein was used as an out-group. The UniProt or NCBI IDs of the proteins used to build the LMO7 phylogenetic tree are presented in Appendix A. Our results revealed that the LMO7 sequences form two subgroups. One encompasses the tetrapod LMO7 and Lmo7a of holostean and teleost fishes while the other includes the lmo7b teleost protein (Figure 1). The common ancestry of the different LMO7 orthologs is further evidenced by the presence of syntenic gene sets around the LMO7 genes, as displayed in Appendix A.

Our data on placental mammals, chicken and *Xenopus* are in accordance with a previous study but differ regarding zebrafish, given that only a single *lmo7* gene was described previously in this species [13]. The presence of two *lmo7* genes in the genome of teleost fishes is expected given that an additional round of whole genome duplication occurred at the base of the teleost fish lineage [18,19]. Although additional analyses are required to determine which teleost species bear both *lmo7* genes or whether there are further unidentified paralogs in specific teleost fishes, our results demonstrate for the first time that *lmo7* underwent duplication in teleosts, generating *lmo7a* and *lmo7b*.

### 2.2. Characterization of the LMO7 Transcripts in Human and Vertebrate Model Organisms

Alternative splicing-derived transcripts of *LMO7* have been described for human [20] and other placental mammals [2]. However, data on *LMO7* variants generated either by alternative splicing or distinct starting sites for transcription or termination are still scarce. Therefore, we analyzed the information about the *LMO7* transcripts available in the Ensembl genome browser for humans and the major vertebrate model organisms (mouse, chicken, *Xenopus* and zebrafish).

Our analysis revealed that, like other genes encoding the PDZ–LIM proteins [21], the *LMO7* ortholog genes are targets of alternative splicing given that several different transcripts were identified in all the species evaluated (Appendix A). In human, there are 27 different predicted *LMO7* transcripts, nine of which are non-coding. Among them, only three feature the highest Ensembl Transcript Support Level (TSL:1, Appendix A). In mice, 16 transcripts were predicted for the *Lmo7* gene (eight with TSL:1, Appendix A), nine of which are non-coding. In chicken and *Xenopus*, eight and six protein-coding *LMO7/lmo7* transcripts were predicted, respectively, although no Ensembl tags were available to assess the prediction reliability of these transcripts (Appendix A). In zebrafish, 13 transcripts are predicted for the *lmo7a* gene (eight are non-coding), while for *lmo7b*, five transcripts are predicted (Appendix A). The exon–intron structure predicted for a well-documented *LMO7* transcript of human and mice as well as the longest transcripts found in the other species evaluated are presented in Figure 2.

Taken together, our findings indicate that studies of expression at the mRNA and protein levels are required to validate the bioinformatic predictions about the *LMO7* variant transcripts in human and in the main vertebrate model organisms. In addition, as multiple non-coding transcripts were predicted to be generated from the vertebrate *LMO7* genes, functional studies to elucidate the role of such transcripts are required. Given that non-coding transcripts can regulate gene expression in different ways [22] and some can generate active micropeptides [23], non-coding *LMO7* transcripts can potentially play important roles that have yet to be described.

### 2.3. Dissecting the Structural Features of the LMO7 Protein

To deepen our comparative analysis, we evaluated the structural features of the LMO7 proteins encoded by the transcripts shown in Figure 2. The human LMO7 (hLMO7; UniProt ID Q8WWI1) contains three well-folded domains, as predicted from the primary structure (The UniProt Consortium, 2021 [24]). Moreover, there is a domain of unknown function (DUF4757) found in two regions: residues 294–382 and residues 650–787, as reported by the PFAM database [25]. There is one three-dimensional structural model for the PDZ domain of LMO7 (PDB 2eaq, residues 1037–1126) (Figure 3A). This X-ray crystallographic structure shows five antiparallel β-strands resembling a β-barrel (also known as the β-finger) next to an α-helical segment. The folding shared by different types of the PDZ domains accounts for their binding to redundant targets. The PDZ domains are widely present in life, from bacteria to mammals, and recognize short linear motifs (SLiMs) in various proteins, contributing to assembly of multicomponent complexes (reviewed in [26]). Close to the N-terminus, there is a calponin homology (CH) domain involved in the interaction with cytoskeletal proteins and signaling. The C-terminal LIM domain is composed of two zinc fingers that function in cytoskeleton reorganization. Collectively, all three domains of LMO7 are involved in protein–protein interactions. However, the predicted domains only cover 16% of the hLMO7 primary structure. Moreover, LMO7 shows nuclear localization, acting on transcriptional activation and differentiation, processes related to proteins containing long regions of intrinsic disorder [27], and none of its globular domains is related to direct nucleic acid binding. Then, we analyzed whether hLMO7 contains intrinsically disordered regions (IDRs), which are highly flexible regions that lack a fixed secondary and/or tertiary structure [28]. Instead, IDRs are best described as an ensemble of conformations, challenging the structure–function relationship since the plasticity of unordered regions enables control of several regulatory pathways [29]. Analysis by means of the charge–hydropathy (CH) plot [30] shows that the chemical composition of the LMO primary structure is similar to that of proteins containing long IDRs (Figure 3C). Prediction of disordered regions by means of a set of seven different algorithms revealed two long C-terminal disordered regions interspaced with the PDZ and LIM domains, respectively, and a short N-terminal disordered region (Figure 4). In addition to PONDR, IUPred, PrDOs and their derivatives, the D2P2 database [31] verifies IDRs by means of the combination with the Spritz algorithms [32] that have been trained on NMR and X-ray crystallography data as well as biophysical characterizations gathered in the Disprot database (available at disprot.org). The results from D2P2 show consensual regions of disorder (75% accordance among the predictors) as follows: residues 321–344 (where the nuclear localization signal is found), 753–929, 938–1042 and 1236–1603. Interestingly, the enrichment in intrinsic disorder is conserved along evolution (Figure 3C and Figure 4). The LMO7 sequences from mouse (*Mus musculus*), chicken (*Gallus gallus*) and zebrafish (*Danio rerio*) shows an enriched disorder content, with the longest disordered segment appearing in the C-terminus (comprising around 400 residues: 1200–1600 for human and mouse, 1100–1500 for chicken and 900–1300 for zebrafish Lmo7b) (Figure 4). To investigate whether the hLMO7 segments could fold into coiled coils, we used ncoils [33] and observed three segments with high propensity: residues 721–776, 1227–1277 and 1347–1384. Coiled coil structures become folded into α-helical segments upon intermolecular interactions and can form supercoils containing up to seven helices wrapped around each other. Since hLMO7 is involved in many protein interactions, coiled coils could orchestrate the assemblage of large protein clusters. The high content of intrinsic disorder in hLMO7 (PONDR-VLXT total score: 52.5%) together with the prediction of coiled coils, which also function as oligomerization domains, prompted us to verify whether hLMO7 presents potential regions that could drive liquid–liquid phase separation (LLPS). Two bioinformatic tools, catGRANULE [34] and PScore [35], predicted a short C-terminal region comprising about 50 residues, with a potential to drive homotypic LLPS (Figure 5). The catGRANULE algorithm predicts condensation based on the following characteristics: primary sequence composition, intrinsic disorder and nucleic acid-binding propensities [34]. According to catGRANULE, hLMO7 showed a strong total score for LLPS (score: 1.024; a score above zero indicates LLPS) (Figure 5). In contrast, the overall PScore, which was developed on the high frequency of planar sp^2^ pi–pi interactions from LLPS “driver” proteins, was only 3.66, below the minimum threshold of 4, which indicates LLPS [35] (Figure 5). Nonetheless, multiple heterotypic interactions with its molecular partners promoted by its three domains and the IDRs could drive the assemblage of condensates. Consequently, it is essential to assess whether hLMO7 can form biomolecular condensates in vivo. 

Regarding other motifs encoded by the hLMO7 sequence, analysis by Phobius [36] predicts a transmembrane motif (TM; residues 32–50) at the extreme N-terminus (Figure 3A). This may contribute to the insertion in the plasma membrane, and perhaps also to insertion in the nuclear membrane, as LMO7 is found in the nuclear membrane during the differentiation of mouse myoblasts [17]. Interestingly, several LIM domain-containing proteins undergo nucleocytoplasmic shuttling under specific conditions. Indeed, LMO7 is a DNA-binding protein that enhances transcription of key genes related to myogenic differentiation [17]. Accordingly, Holaska et al. [8] suggest two nuclear export signals (NES; residues 118–127 and 650–659) in the N-terminus and a C-terminal nuclear localization signal (NLS; residues 1189–11,996) based on bioinformatic prediction tools. However, to the best of our knowledge, the evaluation tools used in their work were not stated. Hence, we also searched for NLS/NES annotated in the database of nuclear transport sequences based on the experimentally proven sequences [37] and found only one motif that may function as a nuclear localization sequence (NLS) in the N-terminus (328-LRKKP-333) for the human protein (Figure 3B). It was reported that LMO7 N-terminus contains a predicted transactivation domain (TAD). Additionally, residues 888–1320 of recombinant hLMO7 seem to be involved in DNA binding, as evidenced by the electrophoresis mobility shift assay (EMSA) [17]. Using the 9aaTAD prediction tool [38], we found that the N-terminus contains three motifs and the C-terminus contains one motif (residues 931–939) with 92% identity to nine-amino-acid transactivation domains (9aaTAD). Interestingly, this propensity for a C-terminal TAD could explain the requirement of residues 888–1320 from hLMO7 to bind myogenic promoters in vitro. In addition, NLSs are found in LMO7 from all the species studied, apart from the b isoform from zebrafish. A perfect match for 9aaTADs was observed for LMO7 from zebrafish (isoform a), chicken and mouse (Figure 3B). Based on a search through the PhosphoSitePlus database [39], we report the post-translational modifications (PTMs) that occur in hLMO7 (Figure 3C). Most phosphorylated residues lie within the IDRs of hLMO7. We speculate that phosphorylation/dephosphorylation of serine 318, serine 322 and tyrosine 323 controls nuclear shuttling.

To identify differences in domain organization between the five protein-coding splicing variants of hLMO7 reported in UniProt (ID Q8WWI1), we performed primary structure alignment by Clustal Omega (Figure 6 and Figure 7A). The alignment showed that isoform 5 does not possess the predicted TM motif, the CH domain or a short-disordered region predicted by seven disorder algorithms, from residues 182 to 285 that precede the NLS (Figure 6 and Figure 7A). Proteins containing a CH domain have been implicated in actin and tubulin binding and are believed to connect cytoskeleton to signaling pathways [40]. Thus, isoform 5 probably lacks the ability to interact with several proteins to transduce signaling via the CH domain.

Isoform 3 lacks residues 359 to 690 predicted to contain two main antiparallel α-helices and four short α-helices together with long disordered regions (Figure 6 and Figure 7B, highlighted in green). Because the function of the region that is missing in isoform 3 is unknown, it is not possible to predict the functional outcome. Interestingly, all five hLMO7 isoforms share the main long disordered regions, the PDZ domain and the predicted NLS (Figure 6 and Figure 7A). Analysis using the I-TASSER structural alignment program (TM-align) [42] showed that the closest structural analog to the hLMO7′s LIM domain is the LIM domain from thyroid receptor interacting protein 6 (TRIP6; PDB 1X61). Superimposition of the 3D structural models from the LIM domain of hLMO7 (obtained by AlphaFold) and TRIP6 (solved by solution nuclear magnetic resonance spectroscopy) showed a root-mean-square deviation (RMSD) of 2.255 Å, indicating a similar fold despite a low level (27%) of sequence identity, as assessed by Clustal Omega. The LIM domain is absent in isoform 4 and is truncated in isoforms 2 and 5 (Figure 6), presumably impacting the protein–protein interactions which the LIM domain orchestrates. Specifically, isoforms 2 and 5 do not contain two key Zn(II) binding sites (Figure 7C, highlighted in blue). Since proper folding requires zinc ion coordination, the absence of binding residues (Figure 6, dark blue rectangles) would abolish LIM-related functions such as recruitment of diverse proteins, including DNA-binding proteins [43].

Additionally, we used AlphaFold and analysis of coevolutionary couplings to obtain structural insights on hLMO7. AlphaFold is a cutting-edge machine-learning approach that predicts protein three-dimensional structural models for the proteome of human and twenty other organisms. Despite this deep-learning method reporting a very low confidence level for the structural definition of IDRs, a significant correlation to disorder predictors has been demonstrated [44]. The predicted Local Distance Difference Test (pLDDT, range from 0 to 100) measures the accuracy of the AlphaFold model to a corresponding structure resolved by structural techniques. Consistent with the analysis by the seven disorder algorithms, hLMO7 is also predicted by AlphaFold to contain multiple domains linked by long disordered regions (Figure 8).

The analysis of evolutionary couplings among the amino acid residues of hLMO7 by the EVcoupling server indicated several significant couplings consistent with globular domains (Figure 8A). Indeed, these match well with the fold predictions by the AlphaFold server and identify the calponin homology, PDZ and LIM domains (Figure 8B). However, it is worth noting that there are many significant evolutionary couplings within the sequence of LMO7 that are in regions with a low score for AlphaFold prediction confidence (less than 70). These regions are predicted to be mostly intrinsically disordered (Figure 4), and we speculate that these intrinsically disordered regions with significant long-range evolutionary couplings might undergo disorder-to-order transitions, depending on the solution composition of specific cellular contexts. It is also interesting that an unknown domain reported by PFAM (DUF4757) exhibits an abundance of evolutionary couplings among residues 634-701 (Figure 8). This domain is also not well-defined in the 3D model of hLMO7 and has intermediate values of the disorder score (mean disorder around 0.5).

### 2.4. Identification of a Putative Regulator of LMO7 Transcription in the Myogenic Context

Given that *LMO7* has been described as having a role during skeletal muscle differentiation of different vertebrate species [8,10], we wondered whether the *LMO7* ortholog genes would be regulated by an evolutionarily conserved element (e.g., a promoter or an enhancer). Therefore, we searched for evolutionarily conserved regions (ECRs) in the *LMO7* loci of human, mouse, chicken, *Xenopus tropicalis* and zebrafish (*lmo7a* loci) using the ECR genome browser.

Our comparative analysis revealed an ECR present in the *LMO7* loci of all species analyzed except for zebrafish. In humans, this conserved element (hLMO7 ECR) is found upstream of the first exon of the Lmo7-207 transcript (ENST00000465261.6; see Appendix A for details). Several potential transcription factor binding sites (TFBS) for proteins involved in skeletal myogenesis, such as PAX3, MYOD, MYOGENIN and MEF2, were identified in the human ECR (Figure 9). In addition, this ECR contains LEF1/TCF4 potential binding sites, indicating that this element may be responsive to WNT ligands, which are known to play multiple and pivotal roles in skeletal muscle development [45]. The MultiTF search for conserved TFBSs among human and other species showed that MEF2 and LEF1/TCF4 sites are evolutionarily conserved.

To further evaluate the regulatory potential of the human hLMO7 ECR, its sequence was used to perform a BLAT search against the human genome (assembly GRCh38/hg38) in the UCSC genome browser. Our analysis revealed that the hLMO7 ECR partially overlaps with a predicted *LMO7* regulatory element of the GeneHancer promoter/enhancer catalog (ID GH13J075758), a database of human regulatory elements and their inferred target genes [46]. Of interest, the GH13J075758 element was also categorized as a super enhancer related to myogenic differentiation in dbSUPER (ID SE_37705), a database of super enhancers in the mouse and human genomes [47].

Overall, our findings indicated that a conserved regulatory element is found in the *LMO7* loci of humans and other vertebrates, that may be involved in regulating gene transcription in the context of skeletal myogenesis. Functional studies are required to confirm the role of the ECR identified here as an *LMO7* regulatory element.

### 2.5. Intracellular Distribution of the LMO7 Protein in Chicken and Zebrafish Embryonic Muscle Cells

Since the PDZ–LIM proteins have been associated with multicompartmental cell functions during development, we decided to explore the cellular distribution of the LMO7 protein during vertebrate skeletal muscle development. To achieve this objective, we analyzed the intracellular localization of LMO7 in two widely used vertebrate animal models for the study of myogenesis. Zebrafish embryos and primary cultures of chicken embryonic muscle cells were labeled for LMO7 and analyzed with a confocal laser microscope. First, we analyzed the localization of Lmo7 in zebrafish somites. Zebrafish embryos are particularly appropriate for studies on skeletal muscle cell development because of their transparency and external development, which allow access to embryos for easy and detailed visualization. Furthermore, in a single zebrafish embryo, it is possible to analyze different differentiation stages of somite progenitor muscle cells. Zebrafish somites form one after another from tissue at the tail end of the embryo so that somites near the tail of the fish are younger and somites near the head are older. Our results showed Lmo7 near the septa and the notochord, and in the cytoplasm of myogenic cells in the somites from the trunk region of prim 25 zebrafish embryos [48] (Figure 10A–C). In somites from the caudal region of prim 25 zebrafish embryos, Lmo7 was found in the cytoplasm and was particularly concentrated in the perinuclear region of myogenic cells (Figure 10D–F). No labeling of Lmo7 was detected within the nuclei of progenitor skeletal muscle cells in zebrafish embryos (Figure 10). The presence of Lmo7 near the septa of zebrafish somites might be related to its role in skeletal muscle cell adhesion to the extracellular matrix (ECM), whereas Lmo7 perinuclear localization might be related to its role in intracellular signaling. Recently, the perinuclear region of eukaryotic cells has been described as a space that concentrates signaling proteins distributed in a 3D network of cytoskeletal filaments and organelles [49].

Next, we analyzed the distribution of LMO7 in chicken myogenic cell cultures under a fluorescence confocal microscope and found LMO7 within the nuclei of mononucleated myoblasts and in the cytoplasm (perinuclear region) of multinucleated myotubes (Figure 11A–P). These results agree with a previous report showing that upon myotube formation in the mouse C2C12 cell line, LMO7 shuttled from the nucleus to the cytoplasm [17]. Furthermore, our data are in accordance with the predicted function of the PDZ–LIM proteins regarding their ability to mediate signals between the nucleus and the cytoplasm [5] since LMO7 was found in the nucleus and in the perinuclear cloud of chicken muscle cells. The presence of LMO7 within the nuclei of chicken myoblasts reinforces the idea that LMO7 has a role in the regulation of gene expression in skeletal muscle cells [8,17]. Since beta-catenin is a major transcriptional regulator in muscle cells [50], we decided to investigate a possible crosstalk between the Wnt/beta-catenin pathway and the LMO7 signaling pathways during chicken myogenesis. To test that, we treated chicken myogenic cells with two activators of the Wnt/beta-catenin pathway, BIO and Wnt3a, and analyzed possible alterations in the intracellular distribution of LMO7. We selected BIO and Wnt3a for these experiments since both molecules have been shown to be robust activators of the canonical Wnt/beta-catenin signaling pathway. BIO is a potent and selective pharmacological inhibitor of glycogen synthase kinase-3β (GSK3-β), and it is well-established that inhibition of GSK3-β allows the nuclear translocation of beta-catenin and the subsequent beta-catenin-dependent regulation of its target genes [51]. Different from BIO, Wnt3a is a member of the canonical Wnt glycoproteins that can bind to its receptor Frizzled (Fz) and coreceptor lipoprotein receptor-related protein (LRP5/6) at the plasma membrane of the target cells and activate the canonical Wnt/beta-catenin signaling pathway. Since these two activators of the Wnt/beta-catenin pathway (BIO and Wnt3a) differ in their mechanisms of action, testing their effects in chicken muscle cells could provide more robust information on the possible interplay between LMO7 and the Wnt/beta-catenin signaling pathways. Interestingly, both BIO and Wnt3a induced an increase in the nuclear labeling of LMO7, which was concentrated in specific compartments within the nuclei of muscle cells (Figure 11, insets M–P). Curiously, LMO7 labeling in fibroblasts was lower than in muscle cells in all the experimental conditions (Figure 11A–P), pointing to a muscle-specific role of LMO7. Quantification of the LMO7 labeling showed an increase in the presence of LMO7 within the nuclei of all the cell phenotypes (myoblasts, myotubes and fibroblasts) after treatment with BIO and Wnt3a (Figure 11Q). We also quantified the amount of LMO7-positive nuclear aggregates in all the experimental conditions and found a significant increase in these aggregates in myoblasts, myotubes and fibroblasts after the activation of the Wnt/beta-catenin pathway (Figure 11R). Since chromatin is not randomly distributed within the interphase nuclei of eukaryotic cells [52], our data suggest that LMO7 could be concentrated at specific nuclear territories with active transcription. Further studies are necessary to identify the specific nuclear domain where LMO7 localizes after the activation of the Wnt signaling pathway in skeletal muscle cells.

Importantly, the LMO7 protein was detected within the nuclei of chicken myoblasts and in the cytoplasm of zebrafish somites (Figure 10 and Figure 11). No labeling of LMO7 was detected within the nuclei of muscle cells in zebrafish embryos (Figure 10). The differences in the intracellular distribution of the LMO7 protein between zebrafish and chicken muscle cells may have different explanations: (i) we analyzed chicken muscle cells grown in vitro as compared with zebrafish embryos grown in vivo. It is possible that the mechanical stress caused by in vitro conditions where the chicken muscle cells were cultivated induced the nuclear translocation of LMO7 and the subsequent activation of target genes. LMO7 has been reported to be associated with focal adhesions (cell–ECM adhesions) in cells by the interaction with p130Cas, a key signaling component of focal adhesions, and that this association allows muscle cells to withstand mechanical stress [9]. Importantly, focal adhesions are rarely seen in vivo, such as in zebrafish embryos, and therefore LMO7 may have a specific role in stress-related responses of muscle cells grown in vitro; and/or (ii) as described above, our results revealed that the Lmo7 teleost genes/proteins are separated into *lmo7a*/Lmo7a and *lmo7b*/Lmo7b. We cannot exclude the possibility that the antibody against Lmo7 that we used in the immunofluorescence experiments with zebrafish embryos was able to detect only one zebrafish Lmo7 protein isoform and that the other undetected isoform could have a different intracellular localization (including muscle cell nuclei in zebrafish somites). More experiments are needed to explore this hypothesis.

### 2.6. A bibliometric Glimpse of All the Published Data on LMO7

Finally, we performed an exploratory analysis of the data retrieved from the PubMed (https://pubmed.ncbi.nlm.nih.gov/) database. Using descriptors “LMO7” OR “Lmo7” OR “LMO-7” OR “Lmo-7” OR “lmo7” OR “lmo-7” OR “Lim domain only protein 7”, the search returned 76 articles as of 10 October 2021 in a period that spanned the years 1998 to 2021. First, we analyzed the number of LMO7 articles published per year and observed that the number of LMO7 publications is increasing over the years, particularly after 2017, which highlights the growing relevance of LMO7 studies (Figure 12). Then, we analyzed the frequency of words from titles and abstracts of the articles using the VOSviewer software [53]. Figure 13 depicts a term map of co-occurrence relations between the scientific terms found in the title and abstract of the 76 LMO7 articles. VOSviewer has its own clustering technique [54], which is based on the citation relations between the clusters. The most frequent words are represented by colored nodes. Five different colors (red, blue, green, yellow and purple) represent five clusters of different scientific contexts for the 76 LMO7 articles (Figure 13). Red represents the central node, where most of the interactions occur, and which is related to studies of LMO7 in cardiac and skeletal muscles (myocardium, Emery–Dreifuss muscular dystrophy). Blue is the second most important node and is associated with the regulation of gene expression (exons, alternative splicing) by LMO7. Green represents studies of the role of LMO7 in cancer cells and organs (oncogene proteins, lung cancer). Yellow is related to the role of LMO7 in cell adhesion (adherens junctions, cadherin). Finally, purple shows the work developed in human genetics of LMO7 (genome-wide association study and single nucleotide polymorphism). The sizes of the nodes denote the citations; that is, the larger the size of the node, the greater the number of citations. The larger nodes were “lim domain proteins”, “humans”, “animals”, “mice” and “tumor cell line”, representing the biological model used in LMO7 studies. Importantly, the large size of the nodes “transcription factors”, “signal transduction” and “cell adhesion” highlights LMO7 versatile functions in the regulation of gene expression, signaling and adhesion processes. The nodes “microfilament proteins” and “cadherins” point to the participation of LMO7 in cadherin/actin-based intercellular adhesion. The collection of these bibliometric analyses shows that LMO7 studies are concentrated in transcriptional regulation, signaling and adhesion in muscles and cancer. Curiously, no node related to intrinsically disordered proteins was detected, reinforcing the novelty of our data.

## 3. Materials and Methods

### 3.1. Bioinformatic Analysis

#### 3.1.1. Identification of the *LMO7* Orthologs Genes

Orthologs of the human *LMO7* gene were identified by textual searches in the Gene NCBI database (https://www.ncbi.nlm.nih.gov/gene/) and Ensembl genome browser (https://www.ensembl.org/index.html). In addition to human, the species evaluated were mouse (*Mus musculus*), opossum (*Monodelphis domestica*), chicken (*Gallus gallus*), zebra finch (*Taeniopygia guttata*), anole lizard (*Anolis carolinensis*), Western clawed frog (*Xenopus tropicalis*), zebrafish (*Danio rerio*), channel catfish (*Ictalurus punctatus*) and spotted gar (*Lepisosteus oculatus*). The spotted gar was included in our analyses as a representative of holostean fishes, a group that separated from teleost fishes before the additional whole genomic duplication that occurred in this lineage [18]. The genes around *LMO7* were also annotated as chromosomal regions displaying conserved synteny are believed to share common ancestry.

#### 3.1.2. Phylogenetic Relationships among the LMO7 Proteins

Multiple alignments of the predicted LMO7 protein amino acid sequences were conducted using ClustalW of Molecular Evolutionary Genetics Analysis (MEGA) software version 11 [55]. Phylogenetic trees were built using the neighbor-joining method of the same software. The robustness of the groupings was assessed using 1000 bootstrap resampling.

#### 3.1.3. *LMO7* Transcripts in Human and Vertebrate Model Organisms

Information about *LMO7* transcripts annotated in human and major vertebrate model organisms (mouse, chicken, *Xenopus tropicalis* and zebrafish) were obtained in the Ensembl genome browser (https://www.ensembl.org/index.html). Transcript flags, which help to identify transcripts with higher quality, were used for selecting the main *LMO7* transcripts for human and mouse. Given that transcript flags are not available for other species analyzed, the longest transcript was chosen for comparisons. Exon–intron DNA sequences were downloaded in the Ensembl genome browser and used to create transcript graphics with online resource Exon-Intron Graphic Maker (http://wormweb.org/exonintron).

#### 3.1.4. Structural Features of LMO7 by Bioinformatics Tools

The intrinsically disordered regions of LMO7 were analyzed by means of the D2P2 database [31] and seven disorder predictors, PONDR-FIT [56], PONDR-VLXT [57], IUPred-long (long regions of intrinsic disorder) and IUPred-short (short regions of intrinsic disorder) [58], PONDR-VSL2 and PONDR-VL3 [59] and PrDOS [60]. The average disorder profile was obtained by calculating the mean of disorder reports from the seven computational tools. A score > 0.5 refers to amino acid residues in disordered regions whereas scores from 0.2 to 0.5 indicate residues in flexible segments. CatGRANULE [34] and PSCore algorithms predicted liquid–liquid phase separation. Coiled coil regions were predicted by ncoils [33]. The Phobius server was used to identify a transmembrane α-helical motif [36]. Nuclear export sequences (NES) and nuclear localization signals (NLS) were analyzed in [37]. Nine-amino-acid transactivation domains (TADs) were analyzed by means of the 9aaTAD prediction tool (available at https://www.med.muni.cz/9aaTAD/index.php) using the moderately stringent pattern and the motifs selected in Figure 3B showed at least 92% match [38]. The 3D structure model prediction was obtained by means of AlphaFold [40]. Coevolution of amino acid residues was analyzed by EVcouplings (available at https://evcouplings.org/) using the default parameters [61].

#### 3.1.5. Identification of the *LMO7* Putative Regulatory Elements

Comparative genomics is a useful tool to identify gene regulatory elements (e.g., promoters or enhancers) [62]. Therefore, we searched for evolutionarily conserved regions (ECRs) among the *LMO7* loci of human, mouse, opossum, chicken, *Xenopus* and zebrafish using the ECR browser (http://ecrbrowser.dcode.org). Human *LMO7* was chosen as base for comparisons with the orthologous sequences. The default parameters of this browser were used in the analyses (ECR = minimum length of 100 bp and 70% sequence identity). The Mulan tool (http://mulan.dcode.org/) was used to access the MultiTF algorithm used to identify conserved transcription factor binding sites [63]. The sequence of the human ECR identified was downloaded and used for a BLAT search in the UCSC genome browser (https://genome.ucsc.edu/) and an analysis using the Regulation tracks.

#### 3.1.6. Bibliometric Analysis

For the bibliometric evaluation, we performed exploratory analyses of the data retrieved from the articles present in the PubMed database (https://pubmed.ncbi.nlm.nih.gov/). The query was performed on 10 October 2021 by using the following descriptors: [lmo7] OR [lmo-7] OR [LMO7] OR [LMO-7] OR [Lmo7] OR [Lmo-7] OR [lim domain only protein 7] OR [Lim Domain Only Protein 7]. We found a total of 76 articles in a period that spanned from 1998 to 2021. We used the freely available software VOSviewer (https://www.vosviewer.com/) to analyze the frequency of words that appear in the titles and abstracts of the 76 articles. A term map of co-occurrence relations between scientific terms was created [53] using the following parameters: all keywords, full counting, minimum number of occurrences of a keyword as 3 (of the 649 keywords, 65 met the threshold), maximum length of circles of 100 and maximum size of lines of 1000.

### 3.2. Analysis of Localization of LMO7 in Zebrafish and Chicken Muscles

#### 3.2.1. Antibodies and Probes

Rabbit polyclonal antibody against LMO7 (code # HPA020923) was purchased from Sigma-Aldrich (St. Louis, MO, USA). DNA-binding probe 4,6-diamino-2-phenylindole dihydrochloride (DAPI) and Alexa Fluor 488-goat anti-rabbit IgG antibody were from Molecular Probes (Eugene, OR, USA).

#### 3.2.2. Chicken Embryonic Skeletal Muscle Cell Cultures

The primary cultures of myogenic cells were prepared from breast muscles of 11-day-old chicken embryos as previously described [64]. Chicken embryos were obtained from Granja Tolomei (Rio de Janeiro, Brazil) and handled according to the Institutional Animal Care and Use Committee protocols under number 069/19. Chicken muscle cells were grown in the minimum essential medium (MEM) with 10% horse serum, 0.5% chicken embryo extract, 1% L-glutamine and 1% penicillin/streptomycin (Invitrogen, São Paulo, Brazil) at an initial density of 7.5 × 10^5^ cells/dish at 37 °C in an atmosphere of 5% CO_2_. For the activation of the Wnt/beta-catenin pathway, 24 h cells were treated for 24 h with 5 µM 6-bromoindirubin-30-oxime (BIO, Sigma-Aldrich) or conditioned media (50% *v/v*) enriched in Wnt3a (obtained from L-Wnt3a cells, ATCC, Manassas, VA, USA).

#### 3.2.3. Zebrafish Husbandry

Zebrafish (*Danio rerio*) were maintained in aquaria with a recirculating water system at 28 ± 1 °C on a 14:10 light/dark cycle in a vivarium localized at the Institute of Biomedical Sciences, Federal University of Rio de Janeiro (Rio de Janeiro, Brazil). The animals were handled and the experiments were carried out according to the Institutional Animal Care and Use Committee protocols under number 039/20. Embryos were collected and rinsed with water three times before chorion removal.

#### 3.2.4. Immunofluorescence Microscopy and Digital Image Processing

Dechorionated zebrafish embryos at 24 hpf were fixed in 4% paraformaldehyde in PBS for 1 h at room temperature (RT). The embryos were then permeabilized with 0.5% Triton-X100 in PBS (PBS/T) for 30 min, blocked in 3% BSA-PBS/T for 1 h and incubated overnight at 4 °C with the primary antibody against Lmo7 (diluted 1:100 in a blocking solution). Chicken myogenic cells were fixed in methanol at −20 °C for 5 min followed by permeabilization in PBS/T for 15 min. The cells were blocked in 50 mM ammonium chloride for 15 min and 3% BSA-PBS/T for 30 min and incubated with the primary antibody against Lmo7 (diluted 1:100 in a blocking solution) for 1 h at 37 °C. Then, the embryos and the cells were washed for 30 min with PBS/T and incubated for 2 h at 37 °C with Alexa Fluor-conjugated secondary antibody (diluted 1:200 in a blocking solution). The nuclei were labeled with 0.1 μg/mL of DAPI in 0.9% NaCl. The embryos and the cells were mounted on #1.5 24 × 60 mm glass coverslips (with spacers in the case of embryos) using Prolong Gold (Molecular Probes). The cells and the embryos were examined under a Leica TCS SPE fluorescence confocal laser scanning microscope (Leica, Wetzlar, Germany). The control experiments with only the secondary antibodies showed only faint background staining (data not shown). Quantification of the LMO7 fluorescence labeling were performed using the Fiji software [65] and figure panels were produced with the Adobe Photoshop software (Adobe Systems Inc., San José, CA, USA), where some of the original fluorescence grayscale images were pseudo-colored and superimposed.

#### 3.2.5. Statistical Analysis

Statistical analysis was carried out using the GraphPad Prism software version 8. The results of three independent experiments are expressed as the means ± standard deviation. Statistical analysis of the data related to the quantification of nucleus versus cytoplasm localization of LMO7 and LMO7-positive nuclear aggregates was performed with two-way ANOVA followed by Tukey’s post hoc test. *p* < 0.05 was considered statistically significant.

## Figures and Tables

**Figure 1 ijms-22-12885-f001:**
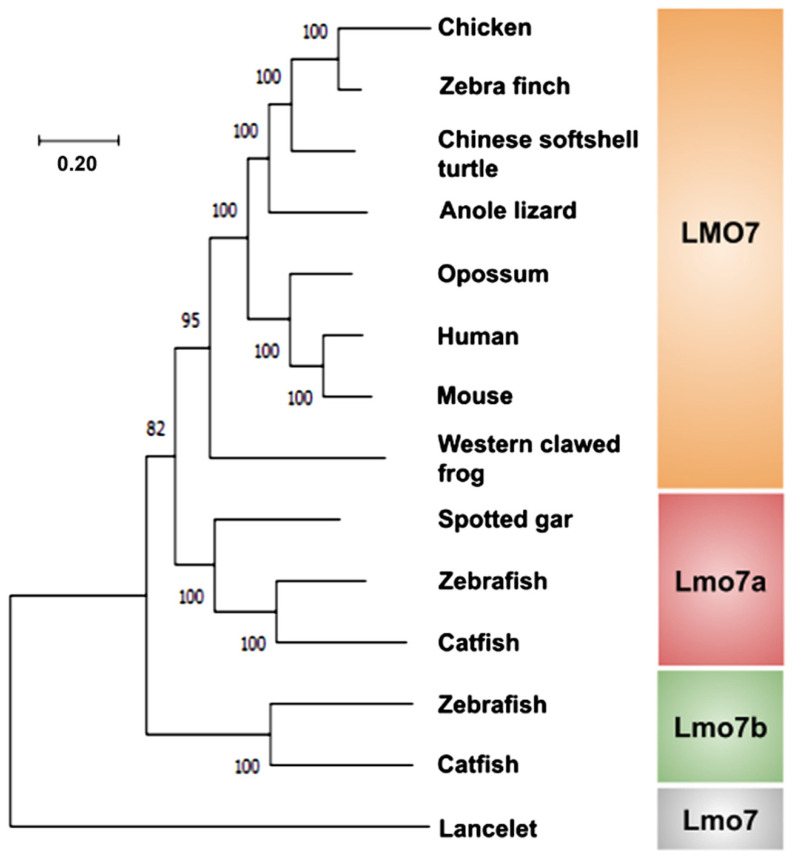
Molecular phylogenetic tree of the LMO7 proteins from representative species of vertebrates. The phylogenetic tree was inferred from amino acid sequences using the *Branchiostoma floridae* Lmo7 protein as an out-group. Bootstrap values were obtained by the neighbor-joining method (1000 replications) using the MEGA 11 software.

**Figure 2 ijms-22-12885-f002:**
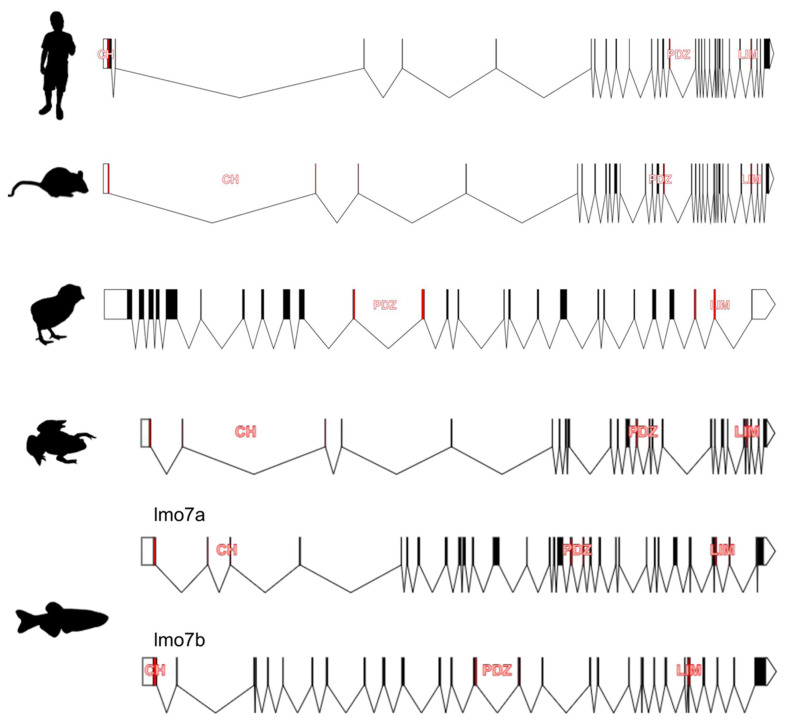
Comparison of the exon–intron organization of the *LMO7* transcripts predicted for human and vertebrate model organisms (mouse, chicken, *Xenopus* and zebrafish). Location of the calponin (CH), PDZ and LIM zinc-binding (LIM) domains is shown for all the species. The CH domain is missing in the chicken *LMO7* transcript. The longest transcripts predicted for the two zebrafish *lmo7* ortholog genes (*lmo7a* and *lmo7b*) are depicted.

**Figure 3 ijms-22-12885-f003:**
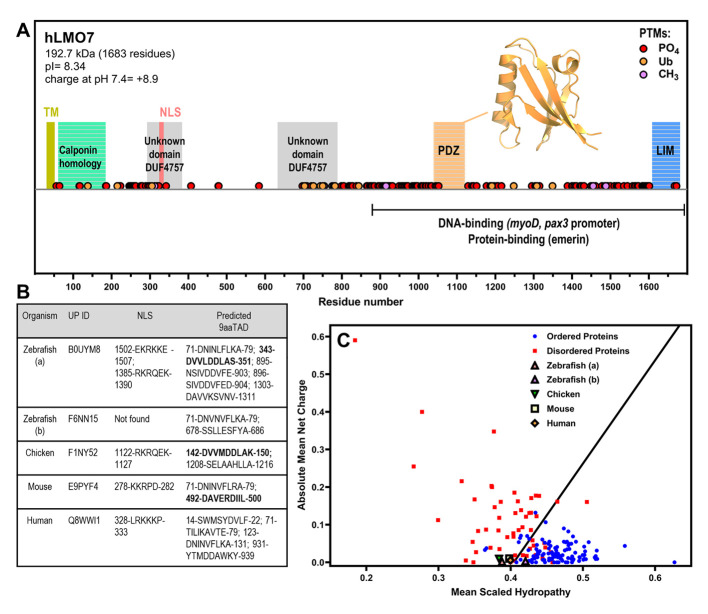
LMO7 is a multidomain protein that has a charge–hydropathy characteristic of proteins enriched in intrinsic disorder and contains nuclear localization sequence(s) and putative nine-amino-acid transactivation motifs. (**A**) Schematic organization of human LMO7 with the domains, motifs and post-translational modifications (PTMs) indicated along sequence. The C-terminal domain mediates interaction with DNA promoters and protein partners. TM, transmembrane α-helix. NLS, nuclear localization sequence. PO4, phosphorylation. Ub, ubiquitination. CH3, methylation. The inset shows the X-ray crystallographic structure of the PDZ domain (PDB 2eaq). (**B**) LMO7 from different species were analyzed by NLSdb (third column, https://rostlab.org/services/nlsdb/, 20 November 2021) and 9aaTAD (fourth column, https://www.med.muni.cz/9aaTAD/). Perfect matches of 9aaTAD are highlighted in bold whereas other annotated motifs correspond to 92% matches. Corresponding UniProt identifiers (UP ID) are shown in the second column. (**C**) Predictor of Natural Disordered Regions (PONDR; http://www.pondr.com/) analysis of charge–hydropathy (CH plot). Well-folded proteins indicated as blue circles and disordered proteins as red squares. LMO7 from the different species is located in the plot region of disordered proteins apart from isoform b of zebrafish.

**Figure 4 ijms-22-12885-f004:**
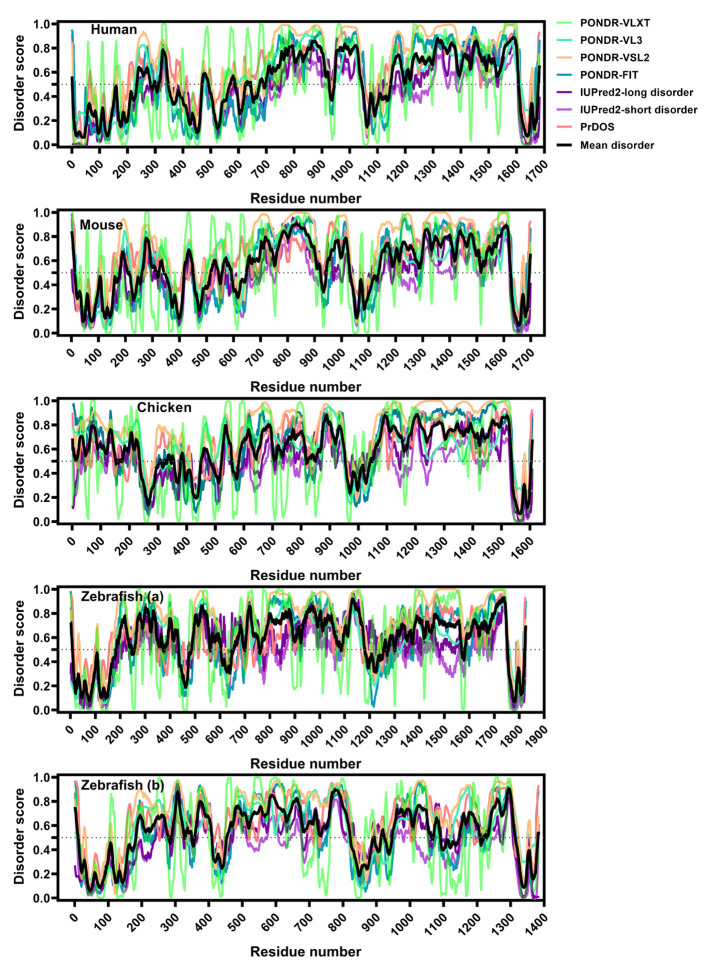
LMO7 is enriched in intrinsic disorder and this feature is conserved across evolution. Analysis of intrinsic disorder by seven disorder predictors along the LMO7 primary structure. Note that the C-terminal protein/DNA-binding region is the longest disordered segment. Data from PONDR-VLXT (light green curve), PONDR-VL3 (green curve) and PONDR-VSL2 (pale orange curve) were provided by http://www.pondr.com/; from PONDR-FIT (blue curve)—by http://original.disprot.org/pondr-fit.php; from IUPred2-long disorder (dark purple curve) and IUPred2-short disorder (purple curve)—by https://iupred2a.elte.hu/plotand; from PrDOS (pink curve)—by http://prdos.hgc.jp/cgi-bin/top.cgi. The average of disorder across all the algorithms was calculated (black curve). Scores higher than 0.5 represent disordered regions, and values between 0.2 and 0.5 indicate flexible segments.

**Figure 5 ijms-22-12885-f005:**
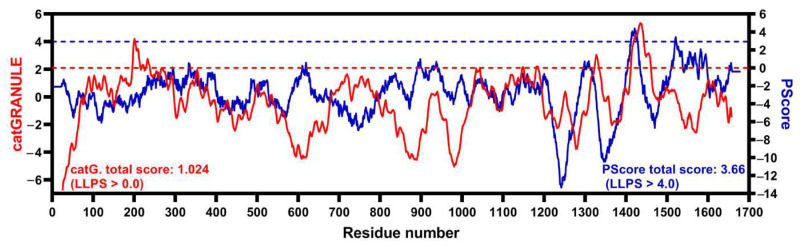
A C-terminal short segment of hLMO7 is predicted to undergo liquid–liquid phase separation. Propensity for hLMO7 LLPS ability along the primary structure calculated by catGRANULE (red curve, http://service.tartaglialab.com/new_submission/catGRANULE) and PScore (blue curve, http://abragam.med.utoronto.ca/~JFKlab/Software/psp.htm).

**Figure 6 ijms-22-12885-f006:**
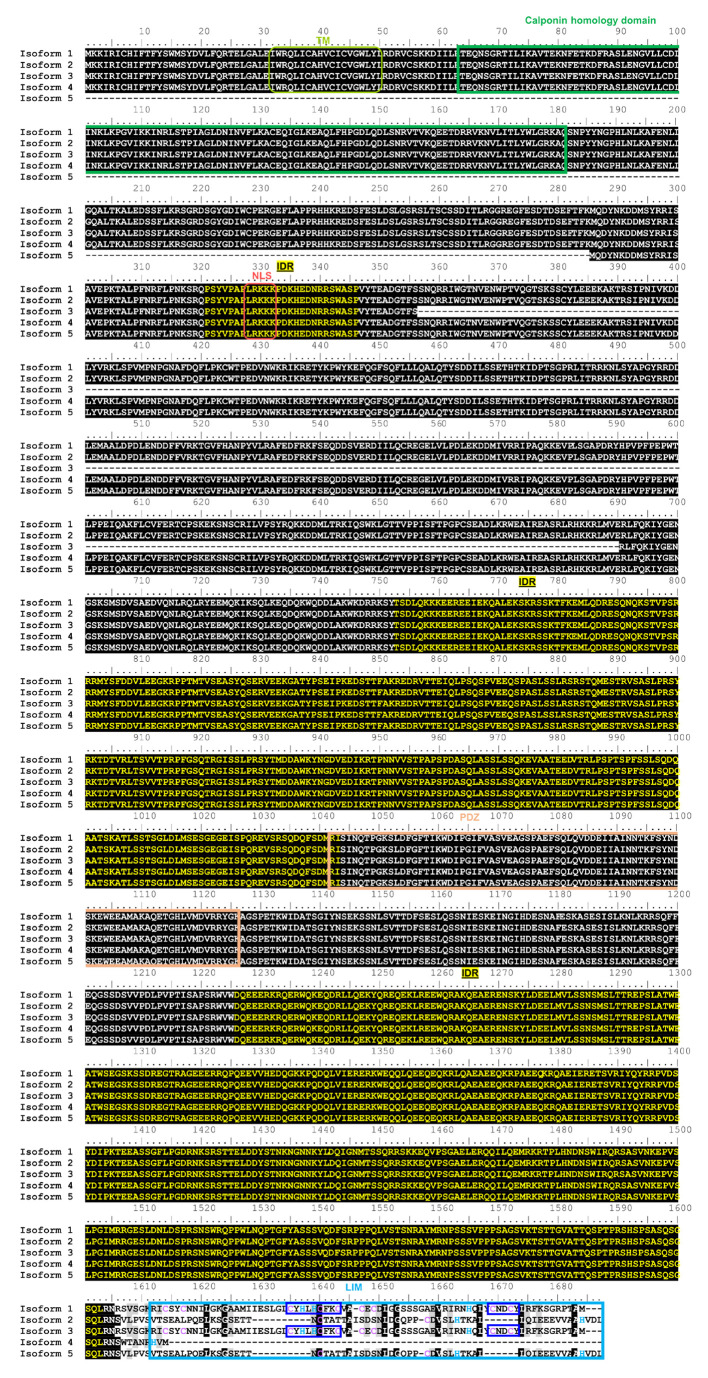
hLMO7 transcript variants show conserved intrinsic disorder but differ in the LIM domain. Alignment of hLMO7 isoforms 1 to 5 (UniProt IDs Q8WWI1-1; Q8WWI1-2; Q8WWI1-3; Q8WWI1-4; Q8WWI1-5, respectively) obtained by Clustal Omega [41]. The predicted domains, motifs and intrinsically disordered regions (IDR) reported in the Database of Disordered Protein Predictions (D2P2) [31] are marked by rectangles and the sequences indicated above. Isoform 5 lacks the putative transmembrane α-helix (TM, predicted by Phobius at https://phobius.sbc.su.se/) and the calponin homology domain (CH). Isoform 3 lacks residues 356–690. The LIM domain is missing in isoform 4. Isoforms 2 and 5 do not have the key zinc ion coordination motifs (dark blue rectangles). Cysteine and histidine residues that might also be involved in Zn(II) binding are marked in purple and blue, respectively. NLS, nuclear localization signal predicted by NLSdb (https://rostlab.org/services/nlsdb/).

**Figure 7 ijms-22-12885-f007:**
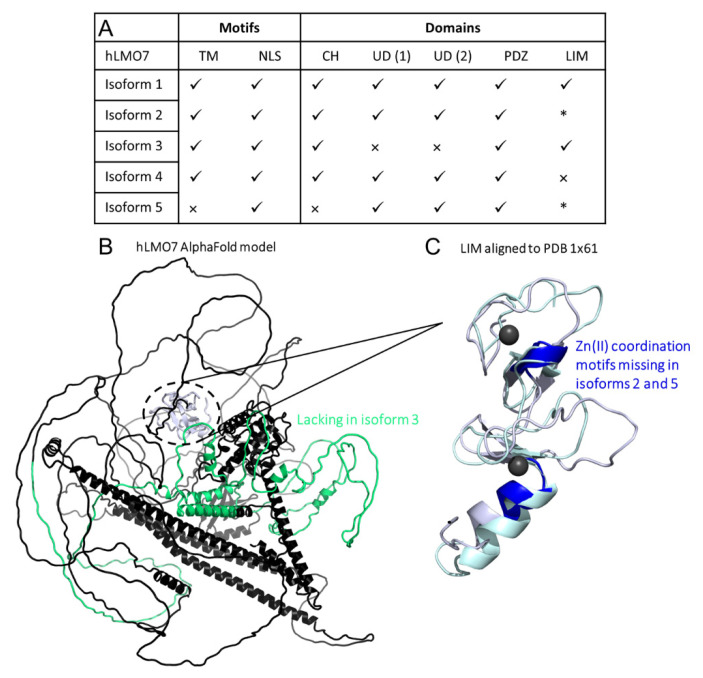
Comparison of domains organization of the five isoforms from human LMO7. (**A**) Summary of results from the primary structure alignment among the hLMO7 isoforms. √, presence. ×, absence. * lacks Zn(II) binding motifs. TM, transmembrane α-helix. NLS, nuclear localization sequence. CH, calponin homology domain. UD, unknown domain DUF4757. (**B**) hLMO7 Alpha Fold-predicted structural model showing the region missing in isoform 3 (green) and the LIM domain (white) which is only conserved in isoforms 1 and 3. (**C**) Superimposition of the LIM domain of hLMO7 (obtained by AlphaFold; white ribbon representation) and the LIM domain of TRIP6 (PDB 1X61; pale green ribbon representation), which is the closest structural analog reported by I-TASSER [42]. Two zinc ions (black spheres) are coordinated in the fingers. Regions highlighted in dark blue are missing in isoforms 2 and 5. Furthermore, isoform 4 does not have the full LIM domain.

**Figure 8 ijms-22-12885-f008:**
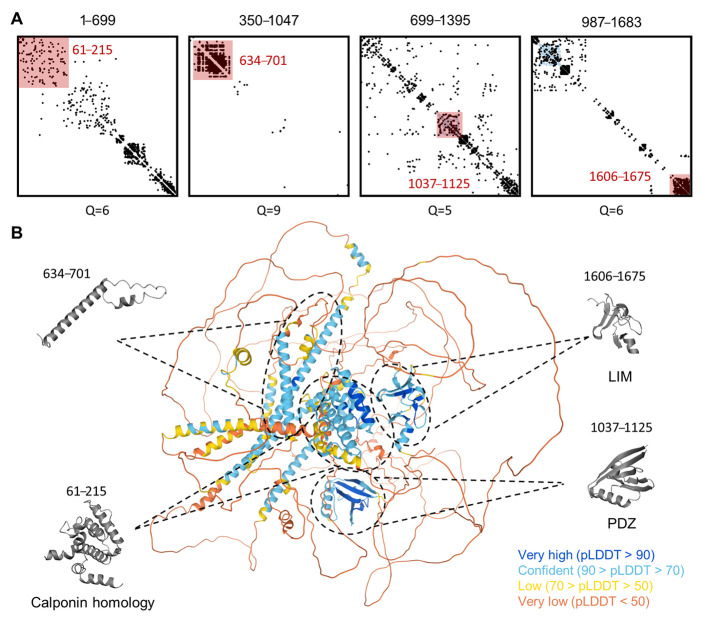
Three-dimensional structure model prediction of hLMO7 indicates the presence of well-folded domains and many intrinsic disorder regions with long-range evolutionary couplings. The evolutionary couplings of overlapping regions of LMO7 were calculated by the EVcoupling server (**A**). The quality scores (Q) for the identification of evolutionary couplings are indicated for each segment and range from 0 (worst) to 10 (best). The pink-shaded squares highlight well-folded domains previously described (residues 61–215, 1037–1125 and 1606–1675) as well as the one identified here (residues 634–701). The 3D structure calculated by the AlphaFold server is indicated in (**B**). Each globular domain highlighted by EVcoupling is indicated in the 3D model. The full-length protein 3D model is colored by the fold prediction confidence score for each amino acid residue (bottom right legend).

**Figure 9 ijms-22-12885-f009:**
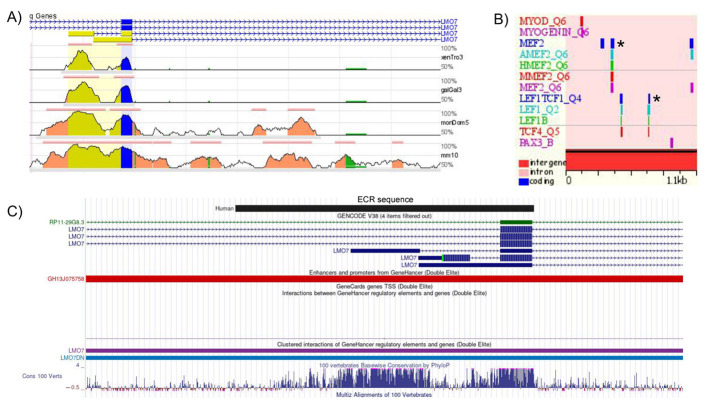
Evolutionarily conserved region (ECR) in the *LMO7* loci of human and other vertebrate species with features of a skeletal muscle-specific regulatory element. (**A**) Conservation profiles of the human *LMO7* genomic region (base sequence, on top) in comparison with the mouse (mm10), opossum (momDom5), chicken (Galgal3) and *Xenopus* (XenTro3) ortholog regions. The 5′–3′ orientation of the human transcripts is denoted by arrow lines. The color codes used to indicate different gene regions are: exons (blue), 5′ UTRs (yellow), introns (pink), repetitive elements (green). Each line represents an alignment, and the vertical height indicates the sequence identity underlying the alignment. (**B**) The human ECR displays several potential binding sites for key regulators of skeletal myogenesis, such as PAX3, MYOD, MYOGENIN and MEF2. Two sets of the LEF1/TCF4 binding sites are predicted in the human ECR. The asterisk indicates the MEF2 and LEF1/TCF4 binding sites that are conserved in mouse, opossum, chicken and *Xenopus.* (**C**) UCSC genome browser display of the BLAT alignment of the human ECR to the human GRCh38/hg38 assembly. The human ECR sequence overlaps with the GH13J075758 promoter/enhancer of the GeneHancer catalog.

**Figure 10 ijms-22-12885-f010:**
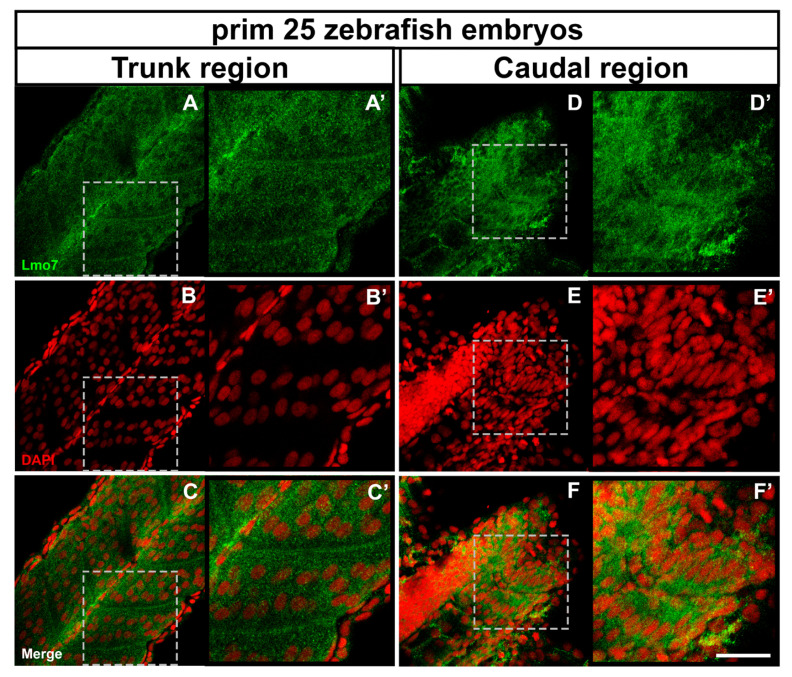
LMO7 distribution in zebrafish embryos. Prim 25 zebrafish embryos were double-labeled with an antibody against LMO7 (green) and the nuclear dye DAPI (red) and analyzed under a confocal laser microscope. The images shown in (**A’**–**F**’) are higher magnifications of the dashed areas marked in the corresponding panels (**A**–**F**). In somites of the trunk region of prim 25 zebrafish embryos, Lmo7 was found near the septa, the notochord and in the cytoplasm of skeletal muscle progenitor cells (**A**–**C** and **A**’–**C**’), while in the caudal region, it was found in the cytoplasm and the perinuclear region of muscle cells (**D**–**F** and **D**’–**F**’). Scale bar = 50 µm.

**Figure 11 ijms-22-12885-f011:**
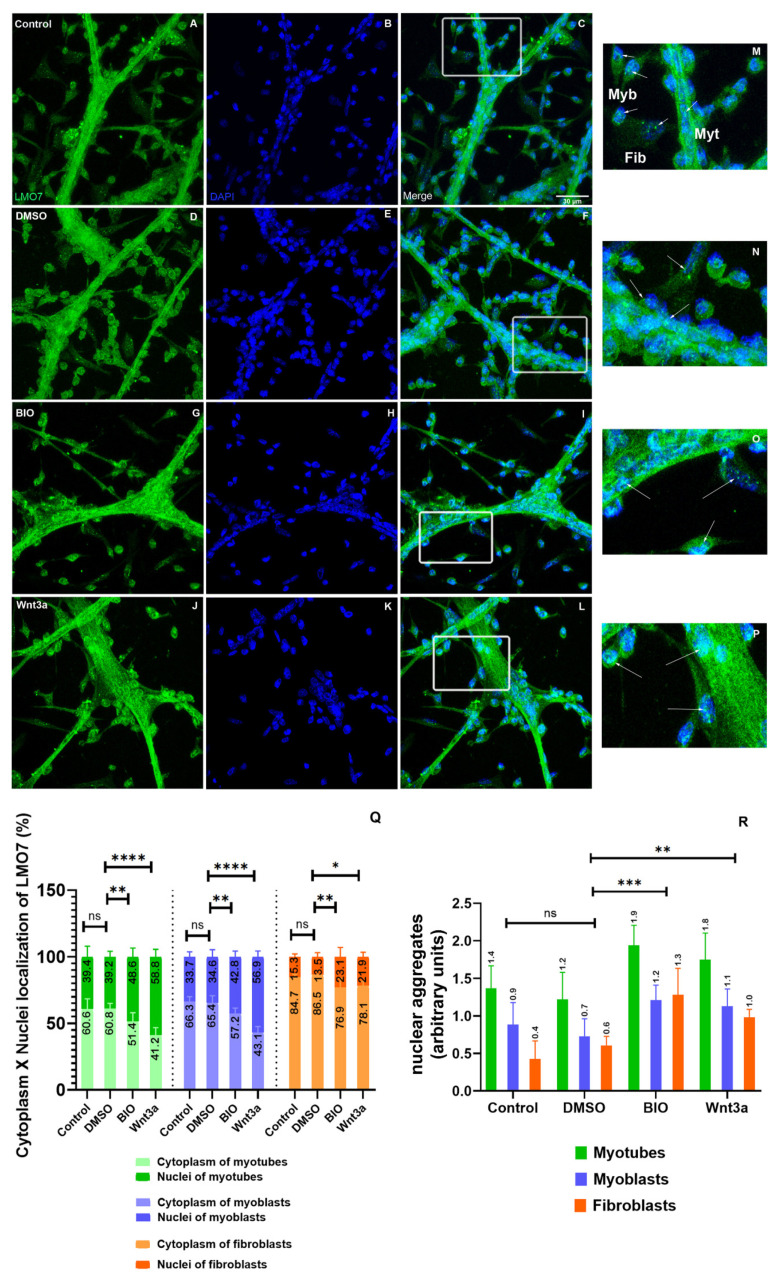
LMO7 intracellular distribution in chicken muscle cells. Chicken myogenic cells were grown for 24 h and treated with DMSO, BIO or Wnt3a for the next 24 h. The cells were double-labeled with an antibody against LMO7 (green) and the nuclear dye DAPI (blue) and analyzed under a confocal laser microscope. Insets (M–P) are higher magnifications of the areas marked in the images **C**,**F**,**I** and **L**. In the control and DMSO-treated cells, LMO7 is found in the perinuclear region of myotubes and within the nuclei of myoblasts (**A**–**F** and **M**,**N**). Both BIO and Wnt3a induced an increase in the nuclear labeling of LMO7 (**G**–**L** and **O**,**P**). Arrows in the insets point to LMO7 localization in myoblasts (Myb), myotubes (Myt) and fibroblasts (Fib). Scale bar for A–L = 20 µm. Quantification of the intracellular localization of LMO7 in the cytoplasm versus the nuclei of cells (**Q**) and of the fluorescence intensity of LMO7-positive nuclear aggregates (**R**) is shown. ns = not significant, * = 0.01, ** = 0.07, *** = 0.004, **** = 0.0001.

**Figure 12 ijms-22-12885-f012:**
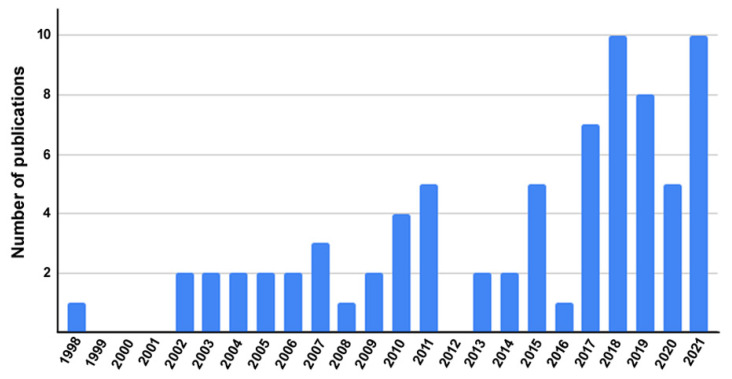
Frequency of LMO7-containing articles published per year. Data retrieved from the PubMed (https://pubmed.ncbi.nlm.nih.gov/) database using descriptors “LMO7” OR “Lmo7” OR “LMO-7” OR “Lmo-7” OR “Lim domain only protein 7”. The search returned 76 articles as of 10 October 2021 for a period that spanned the years 1998 to 2021. From the total of 76 articles, we analyzed the number of LMO7 articles published per year.

**Figure 13 ijms-22-12885-f013:**
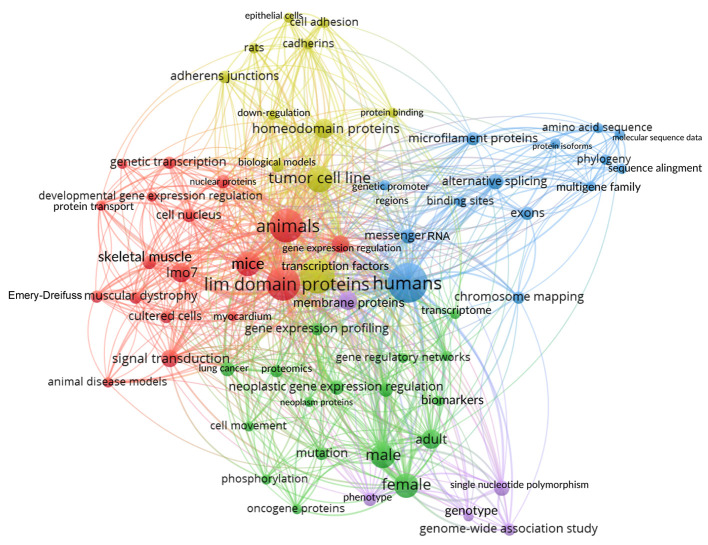
Bibliometric analysis of the co-occurrence relations between scientific terms found in the LMO7 articles. A term map of co-occurrence relations between the scientific terms was created from the bibliometric data retrieved from the titles and abstracts of the published LMO7 articles. The query was performed on October 10, 2021, by using descriptors [lmo7] OR [LMO7] OR [LMO-7] OR [lmo-7] OR [lim domain only protein 7] OR [Lim Domain Only Protein 7]. Five different colors indicate clusters of related terms.

## Data Availability

The authors confirm that the data supporting the findings of this study are available within the article.

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
