# Peer review of "New Findings on LMO7 Transcripts, Proteins and Regulatory Regions in Human and Vertebrate Model Organisms and the Intracellular Distribution in Skeletal Muscle Cells"

_ijms, 2021, doi:10.3390/ijms222312885_

Round 1

Reviewer 1 Report

A brief summary

The authors summarized published data of LMO7, performed in silico analysis of LMO7 and analyzed LMO7 protein localization in myogenic cells. The summary of published data and in silico analysis are well described, however, the experimental results did not support their hypothesis. Further analyses are necessary to prove their hypothesis.

Major comments

  1. Numbering of supplemental Tables and Figures are confusing. There are two Table S2. In addition, Table S3, S4, S5 and S6 are lacking. In the text of Results and Discussion, Figures should be referred in numerical and alphabetical order.

  1. Evolutionary conservation of LMO7 is discussed from Figure 1 to 5. However, LMO7 was localized in cytoplasm in Figure 6 and in nucleus in Figure 7. Is the role of LMO7 not conserved between species? Or is the function different between in vivo and in vitro? The conclusion is not clear. LMO7 localization should be analyzed at different time points in vivo and in vitro.

Author Response

Dear Dr. Tsuchida,

We have submitted the manuscript entitled "New findings on Lmo7 transcripts, proteins and regulatory regions in human and vertebrate model organisms, and its intracellular distribution in muscle cells" by Gomes et al., for consideration for publication in the International Journal of Molecular Sciences. The reviewers recommended a major revision of the manuscript. Therefore, we have made multiple changes in the manuscript text and in its figures to incorporate the suggestions made by the reviewers. In addition, we included a point-by-point response to the reviewer’s comments and concerns below. We would like to thank the reviewers for their careful and meaningful analysis of our manuscript.

Reviewer #1:

1 - The authors summarized published data of LMO7, performed in silico analysis of LMO7 and analyzed LMO7 protein localization in myogenic cells. The summary of published data and in silico analysis are well described, however, the experimental results did not support their hypothesis. Further analyses are necessary to prove their hypothesis. Numbering of supplemental Tables and Figures are confusing. There are two Table S2. In addition, Table S3, S4, S5 and S6 are lacking. In the text of Results and Discussion, Figures should be referred in numerical and alphabetical order.

Author’s response - We apologize for this confusion with the supplementary material. In the current version of our manuscript, we reorganized the supplementary materials and corrected the way figures are referred to in the text (in numerical and alphabetical order).

2 - Evolutionary conservation of LMO7 is discussed from Figure 1 to 5. However, LMO7 was localized in cytoplasm in Figure 6 and in nucleus in Figure 7. Is the role of LMO7 not conserved between species? Or is the function different between in vivo and in vitro? The conclusion is not clear. LMO7 localization should be analyzed at different time points in vivo and in vitro.

Author’s response - We thank the reviewer for this important observation related to the difference in LMO7 distribution between zebrafish and chicken muscle cells. We now have included a new paragraph in the manuscript discussing this point: “Importantly, LMO7 protein was detected within the nuclei of chick myoblasts and in the cytoplasm of zebrafish somites (Figures 10 and 11). No LMO7 labeling was detected in muscle-cell nuclei in zebrafish embryos (Figure 10). This difference in LMO7 intracellular distribution may have different explanations: (i) we analyzed chick muscle cells grown in vitro, as compared to zebrafish embryos grown in vivo. It is possible that the mechanical stress caused by in-vitro conditions where chick muscle cells were cultivated induces the nuclear translocation of LMO7 and the subsequent activation of target genes. LMO7 has been reported to be associated with focal adhesions (cell-ECM adhesions) in cells by the interaction with p130Cas, a key signaling component of focal adhesions, and that this association allows muscle cells to withstand mechanical stress [9]. Importantly, focal adhesions are rarely seen in vivo, such as in zebrafish embryos, and therefore, LMO7 may have a specific role in stress-related responses of muscle cells grown in vitro; and/or (ii) as described above, our results revealed that teleost fishes present two separate lmo7 genes (lmo7a and lmo7b). We cannot exclude the possibility that the antibody against Lmo7 that we used in the immunofluorescence experiments with zebrafish embryos was able to detect only the protein encoded by the zebrafish lmo7a gene and that the protein encoded by lmo7b protein could have a different intracellular localization (including muscle-cell nuclei in zebrafish somites). More experiments are needed to resolve this question”.

Regarding the reviewer’s question about the analysis of LMO7 localization at different time points in vivo and in vitro, we need to point out that in a single zebrafish embryo it is possible to observe all time points of muscle developmental stages. Zebrafish somites form one after another from tissue at the tail end of the embryo, so that somites near the tail of the fish are younger and somites near the head are older. So, we included in the new version of the manuscript the following new text: “First, we analyzed the localization of LMO7 in zebrafish somites. Zebrafish embryos are particularly appropriate for studies on muscle-cell development because of their transparency and external development, which provides embryos for easy and detailed visualization. Furthermore, in a single zebrafish embryo, it is possible to analyze different developmental stages of somite muscle cells. Zebrafish somites form one after another from paraxial mesoderm at the tail end of the embryo, so that somites near the tail of the fish are younger and somites near the head are older. Our results showed LMO7 near the septa and the notochord, and in the cytoplasm of muscle-cell somites from the trunk region of prim 25 zebrafish embryos (Figure 10A-C). In somites from the caudal region of prim 25 zebrafish embryos, LMO7 was found in the cytoplasm and particularly concentrated in the perinuclear region of muscle cells (Figure 10D-F). No labeling of LMO7 was detected within the nuclei of muscle cells in zebrafish embryos (Figure 10)”.

3 - Extensive editing of English language and style required.

Author’s response - The new version of the manuscript was reviewed by a native English speaker.

Reviewer #2:

1 - In the present manuscript, the authors attempted to show the evolution, alternative transcripts, protein structure and gene regulation of LMO7 gene that is an important regulator of myoblast differentiation. Nevertheless, this manuscript can be a type of guide, how to examine in silico important genes based on available public databases. The manuscript is well written. I think that this manuscript is worth publishing in IJMS after minor corrections. In the supplementary file is the mess within tables, please reorganize. The table numbering is finished at table s2 (there is two table s2), and some tables are lacking.

Author’s response - We apologize for this confusion with the supplementary material. In the current version of our manuscript, we reorganized the supplementary materials and corrected the way figures are referred to in the text (in numerical and alphabetical order).

2 - Moreover in the supplementary files is a lack of figure captions.

Author’s response - We apologize for this mistake, and we have now included the figure captions for all the supplementary material.

3 - Line 238-240 – “The alignment showed that isoform 5 do not possess the predicted TM motif, the CH domain and a short-disordered region predicted by the seven disorder algorithms, from residues 182 to 285 located before the NLS” – so which function is missing in this isoform?

Author’s response - We thank the reviewer for this observation. Isoform 5 lacks the calponin-homology (CH) domain, which is implicated in actin cytoskeleton interaction and signaling. We now explain that in the results section in the following statement: “Proteins containing a CH domain have been implicated in actin and tubulin binding and are believed to connect cytoskeleton to signaling pathways (Bañuelos et al, 1998). Thus, isoform 5 probably lacks the ability to interact with several proteins to transduce signaling via the CH domain”.

4 - Line 241-243 – the same as above, which function is missing in isoform 3, due to the lack of certain domains?

Author’s response - To better illustrate the differences regarding motifs/domain organization in human LMO7 isoforms we now include Figure 7. Isoform 3 lacks a region comprising a domain with unknown function reported in PFAM (DUF4757), which is found between residues 294-382 and residues 650-787. Figure 7B shows the region missing in isoform 3 (highlighted in green). Since this region is not yet associated to a known function, it is not possible to ascertain the impact on functional outcome. However, the lack of 6 well-defined secondary-structure elements (shown by the AlphaFold model), namely two long α-helices and four short α-helices as well as disordered regions certainly alters function, pointing towards the need for further investigation in the future. In Figure 7C, we show the Zn(II) coordination regions missing in isoforms 2 and 5 (dark blue) together with alignment between the LIM domain from hLMO7 (modelled by AlphaFold) and the closest structural homolog reported by I-TASSER (LIM domain from TRIP6).

5 - Figure 3 – please add in the figure caption what information is presented by particular curves.

Author’s response - We have updated the legend of the new Figure 4 to indicate the curves. The figure caption now is: “LMO7 is enriched in intrinsic disorder and this is conserved across evolution. Analysis of intrinsic disorder by seven disorder predictors along LMO7 primary structure. Note that the C-terminal protein/DNA-binding region is the longest disordered segment. Data from PONDR-VLXT (light green curve), PONDR-VL3 (green curve), and PONDR-VSL2 (pale orange curve) were provided by http://www.pondr.com/; PONDR-FIT (turquoise curve), by http://original.disprot.org/pondr-fit.php; IUPred2-long disorder (dark purple curve) and IUPred2-short disorder (purple curve), by https://iupred2a.elte.hu/plotand; and PrDOS (pink curve), by http://prdos.hgc.jp/cgi-bin/top.cgi. The average of disorder across all algorithms was calculated (black curve). Scores higher than 0.5 represent disordered regions, and values between 0.2 and 0.5 indicate flexible segments”.

6 - Line 325-336 – please add one sentence of comment, this LMO7 location in somites can be associated with, what kind of LMO7 function.

Author’s response - We now added in the new version of the manuscript the following explanation about the possible functions of LMO7 in zebrafish somites: “The presence of LMO7 near the septa of zebrafish somites may be related to its role in muscle-cell adhesion to the extracellular matrix (ECM), whereas LMO7 perinuclear localization may be related to its role in intracellular signaling. Recently, the perinuclear region of eukaryotic cells has been described as a space that concentrates signaling proteins distributed in a 3D network of cytoskeletal filaments and organelles [44]”.

7 - Line 344-354 – because in the zebrafish LMO7 was not found in the cell nuclei and in the chick, it was found. Please add one sentence of comment from the evolution point of view.

Author’s response - We thank the reviewer for this suggestion related to the differences in the distribution of LMO7 between zebrafish and chicken muscle cells. As explained above for Reviewer #1, we now have included the following new paragraph in the manuscript discussing this point: “Importantly, LMO7 protein was detected within the nuclei of chick myoblasts and in the cytoplasm of zebrafish somites (Figures 10 and 11). No labeling of LMO7 was detected within the nuclei of muscle cells in zebrafish embryos (Figure 10). The differences in the intracellular distribution of LMO7 protein between zebrafish and chick muscle cells may have different explanations: (i) we analyzed chick muscle cells grown in vitro, as compared to zebrafish embryos grown in vivo. It is possible that the mechanical stress caused by in-vitro conditions where chick muscle cells were cultivated induces the nuclear translocation of LMO7 and the subsequent activation of target genes. LMO7 has been reported to be associated with focal adhesions (cell-ECM adhesions) in cells by the interaction with p130Cas, a key signaling component of focal adhesions, and that this association allows muscle cells to withstand mechanical stress [9]. Importantly, focal adhesions are rarely seen in vivo, such as in zebrafish embryos, and therefore, LMO7 may have a specific role in stress-related responses of muscle cells grown in vitro; and/or (ii) as described above, our results revealed that lmo7 teleost sequences are separated into lmo7a and lmo7b (Figure 1), since the teleost underwent an additional round of genomic duplication in relation to the lineage that generated jawed vertebrates [18]. We cannot exclude the possibility that the antibody against Lmo7 that we used in the immunofluorescence experiments with zebrafish embryos was able to detect only one zebrafish Lmo7 isoform and that the other undetected isoform could have a different intracellular localization (including muscle-cell nuclei in zebrafish somites). More experiments are needed to resolve this question”.

8 - Subsection: A bibliometric glimpse of all published data on LMO7 – please add one-sentence comment considering the frequency of occurrence information of LMO7 in recent years, which indicate that this theme is interesting and searched (has increased trend).

Author’s response - We agree with the reviewer that the frequency of occurrence of LMO7 papers is an important information. So, we included in the manuscript a new figure (Figure 12) showing the number of LMO7 papers between 1998 and 2021 and pointing out that there has been an increase in LMO7 publications in recent years. We included in the manuscript text the following: “First, we analyzed the number of LMO7 articles published per year and observed that the number of LMO7 publications is increasing over the years, particularly after 2017, which highlights the growing relevance of LMO7 studies (Figure 12)”.

9 - Line 513-515. Please add here information about the reason for BIO and Wnt3a treatments.

Author’s response - We thank the reviewer for this important comment. Now, we included in the text the following information about the reason to use BIO and Wnt3a in our experiments: “To test that we treated chick myogenic cells with two activators of the Wnt/beta-catenin pathway, BIO and Wnt3a, and analyzed possible alterations in the intracellular distribution of LMO7. We selected BIO and Wnt3a for these experiments since both molecules have been shown to be robust activators of the canonical Wnt/beta-catenin signaling pathway. BIO is a potent and selective pharmacological inhibitor of glycogen synthase kinase-3β (GSK3-β), and it is well established that inhibition of GSK3-β allows the nuclear translocation of beta-catenin and the subsequent beta-catenin dependent-regulation of target genes (Meijer et al., 2003). Different from BIO, Wnt3a is a member of the canonical Wnt glycoproteins which can bind to its receptor Frizzled (Fz) and co-receptor lipoprotein receptor-related protein (LRP5/6) at the plasma membrane of target cells and activates the canonical Wnt/beta-catenin signaling pathway. Since these two activators of the Wnt/beta-catenin pathway (BIO and Wnt3a) differ in their mechanisms of action, testing the effects of both in chick muscle cells could provide more robust information on the possible interplay between LMO7 and Wnt/beta-catenin signaling pathways”.

We hope that the modifications made in the new version of the manuscript have properly addressed the criticism and suggestions made by the reviewers, and that the improvements made in the manuscript will be sufficient for its publication in International Journal of Molecular Sciences.

I would like to state that all listed authors qualify for authorship and agreed in the submission of the revised form of the manuscript. The final version of the manuscript has been seen and approved by all coauthors. The authors declare that they have no conflict of interest.

With kind regards,

Claudia Mermelstein

Reviewer 2 Report

Ijms-1452171 - New findings on Lmo7 transcripts, proteins and regulatory regions in human and vertebrate model organisms, and its intra- 3 cellular distribution in muscle cells

In the present manuscript, the authors attempted to show the evolution, alternative transcripts, protein structure and gene regulation of LMO7 gene that is an important regulator of myoblast differentiation. Nevertheless, this manuscript can be a type of guide, how to examine in silico important genes based on available public databases. The manuscript is well written. I think that this manuscript is worth publishing in IJMS after minor corrections.

Minor comments

In the supplementary file is the mess within tables, please reorganize. The table numbering is finished at table s2 (there is two table s2), and some tables are lacking.

Moreover in the supplementary files is a lack of figure captions.

Line 238-240 – „The alignment showed that isoform 5 do not possess the predicted TM motif, the CH domain and a short-disordered region predicted by the seven disorder algorithms, from residues 182 to 285 located before the NLS” – so which function is missing in this isoform?

Line 241-243 – the same as above, which function is missing in isoform 3, due to the lack of certain domains?

Figure 3 – please add in the  figure caption  what information is presented by particular curves

Line 325-336 – please add one sentence of comment, this LMO7 location in somites can be associated with, what kind of LMO7 function

Line 344-354 – because in the zebrafish LMO7 was not found in the cell nuclei and in the chick it was found. Please add one sentence of comment from the evolution point of view.

Subsection: A bibliometric glimpse of all published data on LMO7 – please add one-sentence comment considering the frequency of occurrence information of LMO7 in recent years, which indicate that this theme is interesting and searched (has increased trend).

Line 513-515. Please add here information about the reason for BIO and Wnt3a treatments

Author Response

(The authors gave the same response as above.)

Round 2

Reviewer 1 Report

The authors revised their manuscript, and  I recommend that the revised manuscript should be accepted for publication.

Author Response

Dear Ms. Talaya Zhu,

We have submitted the revised version of the manuscript entitled "New findings on Lmo7 transcripts, proteins and regulatory regions in human and vertebrate model organisms, and its intracellular distribution in muscle cells" ijms-1452171 by Gomes et al., for consideration for publication in the International Journal of Molecular Sciences. A minor revision of the manuscript was now requested. Therefore, we have made multiple changes in the manuscript text and figures to incorporate the suggestions made by the reviewers and editor. In addition, we included a point-by-point response to the comments and concerns below. We would like to thank the reviewer and editor for their careful and meaningful analysis of our manuscript.

1 - Please correct minor comments below in the proof. Please check whether the authors' description of Lmo7 is correct throughout the manuscript. There are LMO7, Lmo7 and lmo7 in the manuscript. Is it fine to write lmo7 in Lancelet in Figure 7? It is written as Lmo7 in other species.

Author’s response - Different writing styles (LMO7, Lmo7 and lmo7) were used along the manuscript since the rules to denote printed gene and protein symbols differ among species. The nomenclature rules followed by us are summarized for the different vertebrate species in the link (https://en.wikipedia.org/wiki/Gene_nomenclature). These rules are in accordance with the Hugo Gene Nomenclature Committee (HGNC) recommendations for human gene nomenclature (https://www.genenames.org/) and, for the other species, follow specialized committees. For the lancelet, an invertebrate chordate, to the best of our knowledge nomenclature rules are yet to be established. Therefore, we choose to follow the recommendations described for tunicates (also chordate invertebrates), as described in (https://www.ncbi.nlm.nih.gov/pmc/articles/PMC4308547/). A systematic review of the spelling of Lmo7 genes and proteins for the different species was carried out throughout the manuscript and in the Figures.

2 - Please check the protein in Table S6. It is not properly shown in the pdf file that the reviewer sees.

Author’s response - We thank the reviewer for this important comment. We now have corrected Table S6.

3 - A duplicate check has just been done for your manuscript (Similarity: 40%) and we found some overlap sentences (even seven continuous words in one sentence) with previous works in your manuscript which is not allowed by us. So, we kindly suggest you to rewrite them (especially the highlighted content is Introduction) and lower significantly the similarity index when you make the revisions.

Author’s response - We apologize for the high overlap in sentences, which we believe are related to previous papers published by our own research group. In the current version of our manuscript, several sentences from the Introduction and other parts of the manuscript text were reformulated in order to reduce the similarity with previously published articles.

We hope that the modifications made in the new version of the manuscript have properly addressed the criticism and suggestions made by the reviewers, and that the improvements made in the manuscript will be sufficient for its publication in the International Journal of Molecular Sciences.

I would like to state that all listed authors qualify for authorship and agreed in the submission of the revised form of the manuscript. The final version of the manuscript has been seen and approved by all coauthors. The authors declare that they have no conflict of interest.

With kind regards,

Prof. Claudia Mermelstein